# Electron cascade for distant spin readout

Cornelis J. van Diepen [1], Tzu-Kan Hsiao[1], Uditendu Mukhopadhyay[1], Christian Reichl[2], Werner Wegscheider[2] & Lieven M. K. Vandersypen [1]✉

The spin of a single electron in a semiconductor quantum dot provides a well-controlled and long-lived qubit implementation. The electron charge in turn allows control of the position of individual electrons in a quantum dot array, and enables charge sensors to probe the charge configuration. Here we show that the Coulomb repulsion allows an initial charge transition to induce subsequent charge transitions, inducing a cascade of electron hops, like toppling dominoes. A cascade can transmit information along a quantum dot array over a distance that extends by far the effect of the direct Coulomb repulsion. We demonstrate that a cascade of electrons can be combined with Pauli spin blockade to read out distant spins and show results with potential for high fidelity using a remote charge sensor in a quadruple quantum dot device. We implement and analyse several operating modes for cascades and analyse their scaling behaviour. We also discuss the application of cascade-based spin readout to densely-packed two-dimensional quantum dot arrays with charge sensors placed at the periphery. The high connectivity of such arrays greatly improves the capabilities of quantum dot systems for quantum computation and simulation.

[1] QuTech and Kavli Institute of Nanoscience, Delft University of Technology, 2600 GA Delft, The Netherlands. [2] Solid State Physics Laboratory, ETH Zürich, 8093 Zürich, Switzerland. ✉email: l.m.k.vandersypen@tudelft.nl

Fault-tolerant quantum computation benefits from high connectivity, and requires fast and high-fidelity readout[1]. Qubit connectivity and density are severely limited when charge sensors need to be placed near all quantum dots in the qubit array. Not only the charge sensors themselves take space, but in addition, they require a nearby electron reservoir, which takes even more space. Several proposals for quantum processors based on gate-defined quantum dots, suggest gate-based readout of two-dimensional arrays to overcome this limitation[2–5]. The comparatively low signal-to-noise ratio (SNR) of this approach has hindered reaching the fidelity required for fault-tolerant quantum computation[6–9]. Signal enhancement has been achieved with a latching scheme[10–12], but does not enable the readout of dots far from the sensor. Readout based on shuttling[13] requires paths of empty dots to avoid that qubits are lost into the reservoirs, and the long-distance movement of electrons breaks qubit connectivity. Alternatively, readout via a sequence of swap operations is limited by the product of all the swap fidelities[14,15].

We show that charge information can be transferred along a quantum dot array with a cascade, in which the spin-dependent movement of one electron induces the subsequent movement of other electrons. Cascades are used in various fields and technologies: stimulated emission[16] in lasers, secondary emission[17] in photomultiplier tubes, impact ionisation in avalanche photodiodes[18], and neutron-induced decay in nuclear fission[19]. A cascade has also been used to build classical logic with molecules in scanning-tunnelling microscopes[20] and with excess electrons in cellular automata based on Al islands[21,22].

## Results

**Device and cascade concept.** The prototype for cascade-based readout with quantum dots consists of a quadruple dot and a sensing dot. A scanning electron micrograph image of a device similar to the one used in the experiment is shown in Fig. 1a. The device is operated at 45 mK and without an external magnetic field, unless specified otherwise. By applying voltages on the electrodes on the surface we shape the potential landscape in a two-dimensional electron gas (2DEG) 90 nm below, formed in a silicon-doped GaAs/AlGaAs heterostructure. The plunger gates, labelled with $P_i$, control the electrochemical potentials of the dots, and the barrier gates control the tunnel couplings between dots or between a dot and a reservoir.

Figure 1b schematically illustrates the cascade-based readout concept. The first step of the protocol is to perform spin-to-charge conversion, based on Pauli spin blockade (PSB)[23], which induces an initial charge transition conditional on the spin state

of the two leftmost electrons. This transition induces a chain reaction of charge transitions with a final charge transition nearby the sensor, which results in a large change in sensor signal. Resetting the cascade can be achieved by undoing the initial charge transition.

**Quantum dot tuning and charge-stability diagrams.** Figure 2a shows a charge-stability diagram with transitions for the two dots on the left. Unless specified differently, the sensing dot is operated on the low-voltage flank of a Coulomb peak throughout this work. For the tuning and measurements, virtual plunger gates $\widetilde{P}_i$ were used for electrochemical potentials[24–26] and virtual barrier gates for tunnel couplings[24,27,28]. The charge occupation of the four dots is indicated by the numbers in round brackets. The voltages were swept rapidly from right to left (left to right in panel c) and slowly from bottom to top. With these sweep directions, a white trapezoid caused by PSB is visible to the top-left of the inter-dot transition in the charge-stability diagram, with the sensor signal in between the signal for the (1100) and (0200) charge regions. The trapezoid is the region suited for PSB readout. The distance between the inter-dot line, which is the base of the trapezoid, and the top of the trapezoid, corresponds to the singlet-triplet energy splitting.

Cascade Pauli spin blockade (CPSB) is seen in the charge-stability diagram of Fig. 2b. The fourth dot is tuned close to a charge transition, such that the movement of an electron on the left pair induces a change in charge occupation of the fourth dot. See Supplementary Note 1 for details on the tuning of the fourth dot and the sensor for the different charge occupations. Supplementary Note 2 contains an analysis of the anti-crossing sizes.

The charge-stability diagram in Fig. 2c shows both the charge states for PSB and for CPSB readout. This diagram is obtained by varying the detuning of the left pair and the potential of the fourth dot. In this diagram there are three different regimes in $\Delta\widetilde{P}_4$. The left and right regions, with charge transitions indicated with a dashed line, can be used for PSB, with the fourth dot unoccupied and occupied, respectively. The middle region, with a charge transition indicated with a dotted line, can be used for CPSB.

The tuning requirements of the dot potentials for CPSB readout can be further understood from the ladder diagram in Fig. 2d. Dot 4 needs to be tuned such that $\mu_4(1101) < 0 < \mu_4(0201)$, with the electrochemical potential defined as $\mu_i(\ldots, N_i, \ldots) = E(\ldots, N_i, \ldots) - E(\ldots, N_i - 1, \ldots)$, where $E$ is

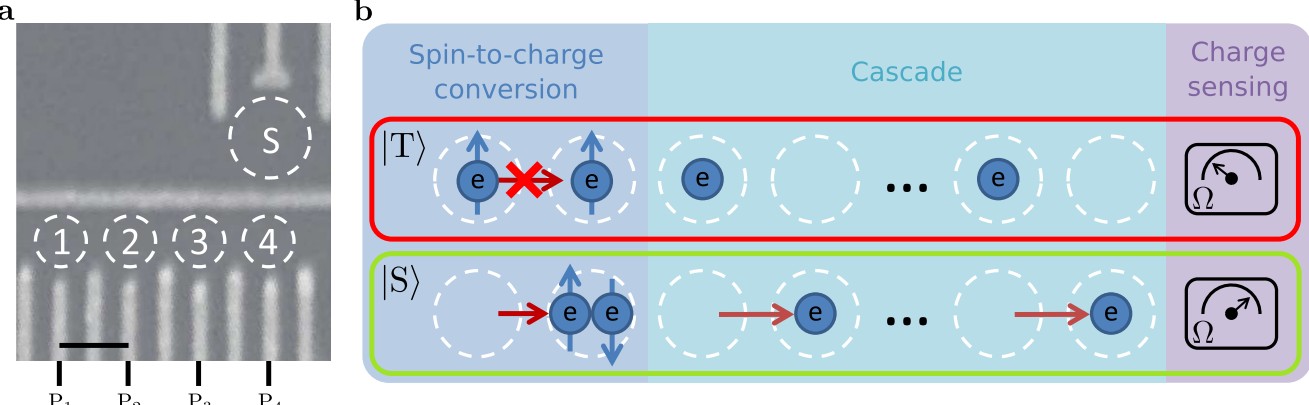

**Fig. 1 Device and cascade concept. a** Scanning electron micrograph of a device nominally identical to the one used for the experiments. Dashed circles labelled with numbers indicate quantum dots in the array, and the dashed circle labelled with 'S' is the sensing dot. The scale bar corresponds to 160 nm. **b** Schematic illustration of spin-to-charge conversion combined with a cascade for electron spin readout on dots far from the charge sensor.

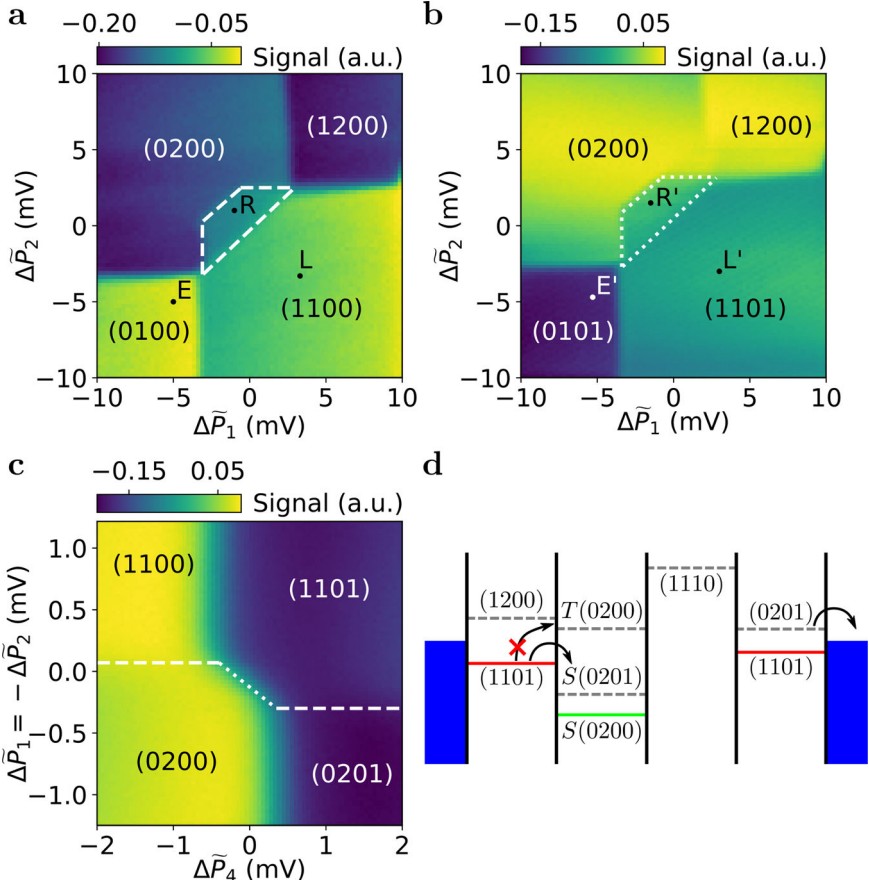

**Fig. 2 Quantum dot tuning for cascade-based readout.** Numbers in round brackets indicate charge occupations of the dots. **a** Charge-stability diagram with transitions for dots 1 and 2. The white, dashed trapezoid on the top-left side of the inter-dot is the Pauli spin blockade (PSB) readout region. The black dots indicate the voltages for the PSB readout cycle: E(mpty), L(oad) and R(ead). **b** Charge-stability diagram similar to **a**, but with different occupations of the rightmost dot. The white, dotted trapezoid is the cascade Pauli spin blockade (CPSB) readout region. White and black dots labelled with E′, L′ and R′ indicate the pulse positions for the CPSB readout cycle. Note that the voltages at the origin are different from those in **a**. **c** Charge-stability diagram showing both the charge states for PSB and CPSB readout. Dashed (dotted) lines correspond to the charge transitions for the PSB (CPSB) readout regions. **d** Ladder diagram illustrating the alignment of the dot electrochemical potentials for the CPSB at the readout point. For a triplet state, the system remains in (1101) (red), whereas for a singlet state it transitions to (0201) and then (0200) (green).

the energy, $N_i$ is the number of electrons on dot $i$, and the Fermi level in the reservoirs is by convention set to zero. This level alignment corresponds to the middle region in Fig. 2c, whereas the left (right) region corresponds to $\mu_4(1101)$ and $\mu_4(0201)$ both above (below) the Fermi level. Note that if $\mu_{2,s}(0201)$ is above $\mu_1(1101)$, the cascade in CPSB readout involves a co-tunnel process (see Supplementary Note 5). In an alternative implementation, we also perform CPSB readout with a charge transition between (1110) and (0201) (see data in Supplementary Note 4).

**Single-shot readout and fidelity analysis.** For single-shot PSB readout, voltage pulses are applied as indicated by the black circles in Fig. 2a. The pulse sequence starts in point E, where the charge occupation is (0100). Then the voltages are pulsed to point L, where an electron is loaded from the reservoir onto the leftmost dot reaching the (1100) charge occupation with random spin configuration. Finally, the voltages are pulsed to the readout point, R, where Pauli spin blockade forces a triplet to remain in the (1100) charge occupation, whereas the singlet transitions to the (0200) charge occupation.

In Fig. 3a the results of 10,000 single-shot measurements are shown in a histogram. The integration time is $t_{int} = 1.5\,\mu s$. The peak at lower sensor signal corresponds to the (0200) charge

occupation, and is assigned as singlet, whereas the peak at higher sensor signal corresponds to the (1100) charge occupation, which is assigned as triplet. Residual overlap between the singlet and triplet distributions induces errors in the distinction of the two charge states, resulting in errors in the spin readout. The inset shows the signal averaged over the single-shot measurements as a function of the time stamp of the integration window. From an exponential fit, the relaxation time, $T_1 = 724\ (70)\,\mu s$, is obtained (see Supplementary Note 6).

For CPSB readout, a pulse cycle similar to that for PSB is used. The sensing dot is operated with comparable sensitivity as for PSB readout. The pulse voltages are indicated with white and black circles in Fig. 2b. The pulse sequence again consists of empty, E′, load, L′, and readout, R′. For CPSB, the charge occupation in E′ is (0101), and in L′, again an electron is loaded on the left dot forming the charge state (1101) with a random spin configuration. At the readout point, owing to Pauli spin blockade, the two electrons on the left remain on separate dots if they are in a triplet state, which results in the charge state (1101). When the two electrons on the left form a singlet state the resulting charge state will be (0200), because the electron on the left dot moves one dot to the right, and the electron on the fourth dot is pushed off due to the cascade effect (here $\mu_1(1101) > \mu_{2,s}(0201)$), so the two charge transitions can occur sequentially, as discussed above).

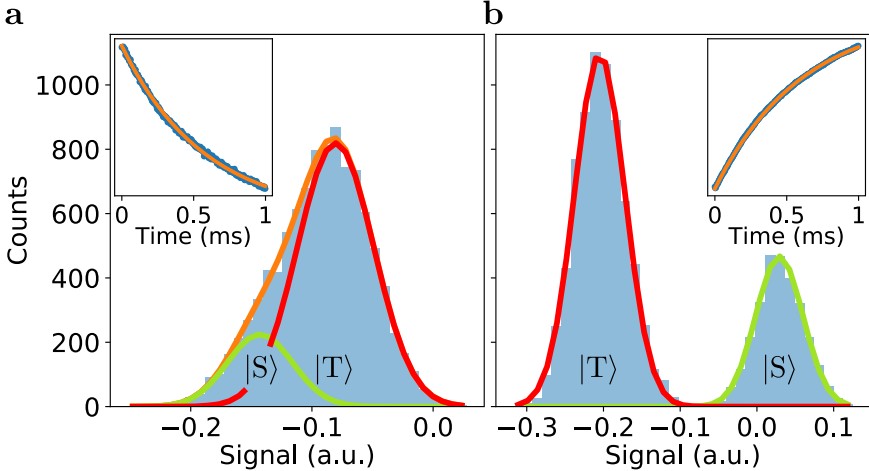

**Fig. 3 Single-shot readout.** Histograms and fits of 10,000 single-shot measurements for **a** PSB readout and **b** CPSB readout. The integration time is $t_{int} = 1.5$ μs. The orange lines are fits to the histograms[9,23] and red and green lines correspond to respectively the triplet and singlet probability distributions. The left (right) inset shows the signal in arbitrary units averaged over the PSB (CPSB) single-shots as a function of wait time in the readout point, and an exponential fit to the data. **a** For PSB readout, the singlet corresponds to charge occupation (0200) and the triplet to (1100). **b** For CPSB readout, the singlet also corresponds to (0200) but the triplet corresponds to (1101), thus with an electron on dot 4.

Figure 3b shows a histogram of 10,000 CPSB single-shot measurements. The integration time, 1.5 μs, is the same as for the PSB single-shot data. The peak at lower sensor signal corresponds to the (1101) charge state, and is assigned as triplet, whereas the peak at higher sensor signal corresponds to the (0200) charge state, which is assigned as singlet. The residual overlap between the singlet and triplet distributions is strongly reduced for CPSB as compared with PSB. Again, from an exponential fit to the averaged single-shot measurements (inset Fig. 3b), the relaxation time, $T_1 = 680$ (3) μs, is obtained.

The cascade enhances the SNR for distinguishing between the singlet and triplet states by a factor of 3.5, extracted by comparing the histogram of CPSB to that of PSB. The SNR is defined as $|V_T - V_S|/\bar{\sigma}_{FWHM}$, with $V_T$ and $V_S$ the signals for a triplet and singlet state, respectively, and $\bar{\sigma}_{FWHM}$ the average of the full width at half maximum of the singlet and the triplet probability distributions. Furthermore, Fig. 3 shows that for PSB the charge signal for the singlet is lower than that for a triplet, whereas for CPSB the charge signal for a singlet is actually higher than for a triplet. We note that changes in screening or shifts in dot positions, do not explain the sensor signals for the different charge occupations.

The enhanced SNR for CPSB readout arises from two contributions. The first contribution is directly due to the cascade, which maps a charge transition far from the sensor to a charge transition nearby the sensor. The longer the cascade, the larger the relative difference, because the final charge transition remains close to the sensor, whereas the initial transition is further away for a longer cascade, thus inducing a weaker sensor signal. The second contribution to the SNR enhancement is because the initial charge transition is an inter-dot transition, whereas the final transition induced by the cascade is a dot-reservoir transition, which has a stronger influence on the sensor.

As for which spin state produces the highest charge signal, for the case of PSB the singlet signal corresponds to a charge moving closer to the charge sensor, thus the sensor signal goes down. For CPSB, a singlet outcome also causes a charge to move closer to the charge sensor, but on top of that, a charge is pushed out of the fourth dot, reducing the total charge on the dot array and removing a charge, which was very close to the sensor. In this case, the two contributions to the signal partially cancel each other, but the resulting effect on the charge sensor is still stronger

for CPSB than for conventional PSB. In Supplementary Note 4 CPSB is implemented such that the charge transition induced by the cascade corresponds to an electron moving closer to the sensor, by having an electron move from dot 3 to dot 4. In this case, the signal was enhanced by a factor of 3.1 as compared with PSB. Here, the effects on the charge sensor of the initial and the final charge transitions add up, but there is no second contribution to the signal enhancement as there is not a mapping of an inter-dot transition to a dot-reservoir transition.

The average spin-readout fidelity with CPSB is potentially above 99.9%, and is achieved within the 1.7 μs readout time. The fidelity for conventional PSB with the same integration time is 85.6%. These fidelities are obtained by analysing different error sources: residual overlap, relaxation, and excitation, but do not include errors in the mapping from the (1101) spin states to the measurement basis at the readout point. Here, we provide details on the analysis for CPSB (see Supplementary Note 7 for details on PSB). The residual overlap between the charge signals and relaxation events during the integration time result in an error of $\eta_{hist} = 0.068\%$ for the average readout fidelity[9,23], as determined from the fit to the single-shot histogram (See Supplementary Note 3). Relaxation during the arming time, $t_{arm} = 0.2$ μs, contributes an error of 0.015%. During the arming time, which is the time between the start of the readout pulse and the start of the integration window, the signal is not analysed as it is still rising due to the limited measurement bandwidth. Excitation during the arming and integration time causes an error in the average readout fidelity of 0.014%, with the excitation time, $T_{exc} = 6.0(3)$ ms (see Supplementary Note 6). The spin readout fidelity will be affected by mapping errors, which can be caused by fluctuations in the hyperfine field (see Supplementary Note 8), high-frequent charge noise, leakage states, and relaxation and excitation during the voltage ramp. The spin readout fidelity including mapping errors can be obtained experimentally by performing high-fidelity deterministic state preparation. Supplementary Note 5 provides an analysis on scaling of the cascade.

## Discussion

A cascade can be implemented in various quantum dot array layouts. Figure 4a shows a schematic illustration of an example of cascade-based spin readout in a two-dimensional array. The

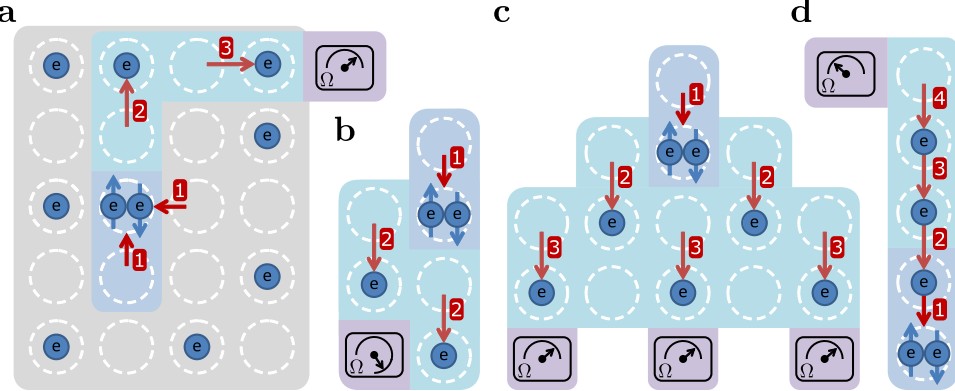

**Fig. 4 Cascade-based readout in 2D, fanout and a dense array.** Schematic illustrations of cascade-based readout **a** in a two-dimensional quantum dot array, **b** using fanout of cascade paths that converge on a single sensor, **c** for fanout with multiple sensors, and **d** in a one-dimensional array with each dot occupied by an electron and where the electrons move away from the sensor. Coloured regions indicate the different aspects of the cascade-based spin readout, with the same colour coding as in Fig. 1b. Numbers indicate the order of the transitions. In **d**, first PSB is performed, then the cascade is activated.

quantum dots are filled in a chequerboard manner, compatible with the proposal in ref. [4], and the sensor is placed at the periphery of the two-dimensional array, with sufficient space for reservoirs. The cascade is implemented by forming a path of dots, which are each tuned close to a charge transition, whereas the dots outside the cascade path are tuned deep in Coulomb blockade so their occupations are unchanged. By tuning different cascade paths the same sensor can be used for the readout of spins at different locations in the quantum dot array. Cascades can also be designed in a fanout shape, as schematically illustrated in Fig. 4b. The ends of multiple cascade paths, both triggered by the same initial spin-dependent charge transition, converge at the same sensor, thus increasing the change in charge distribution in the vicinity of the sensor, which increases the SNR and the readout fidelity. Figure 4c shows another example of fanout of cascade paths, with at the end of each path a sensor. The signal from multiple sensors can be combined to achieve higher SNR and increased readout fidelity. The cascade-based readout can also be performed at higher filling, as schematically illustrated in Fig. 4d, in which each dot is occupied with an electron. The dot emptied by PSB enables the electrons in the cascade to move, which results in a charge moving away from the sensor. In this implementation, the voltage pulse for PSB is first performed, and subsequently, a voltage pulse is sent to tune the dot array to activate the cascade. This two-step procedure suppresses co-tunnelling from (111…) to (210…) without occupying (201…), which can be further suppressed by pulses that lower the relevant tunnel couplings. Accidental dot-reservoir transitions, particularly relevant for cascades at higher filling (see Supplementary Note 2), can be prevented by operating in an isolated regime[29,30].

We end with a few important considerations on the usefulness of the cascade mechanism for spin readout. First, each electron along a cascade path can itself still be operated as a spin qubit, because phase shifts and reproducible (artificial or natural) spin-orbit induced rotations owing to the motion of the electrons can be accounted for in hardware or software[31]. Second, as the length of the cascade path increases, both the spin readout fidelity and the timing of the motion of the electrons, can be largely maintained by allowing a cascade to propagate step-by-step using a series of voltage pulses applied to successive dots, see Supplementary Note 5. Third, the increased SNR from cascades, and the option of further increases through fanout, may enable high-fidelity readout with sensing dots at elevated temperatures[30,32], because the enhanced signal compensates for the additional thermal noise, allowing higher cooling power and integration with cryogenic control or readout electronics[2]. Fourth, the

cascade can also be performed with other spin-to-charge conversion methods, for example with energy-selective tunnelling[33]. Such readout with a cascade does not require a charge sensor nearby the spin to readout, but it does require a nearby reservoir for the initial charge transition. Last, when a quantum dot array is operated with cascade-based readout for qubits then extra caution should be taken for handling correlated errors as these may occur due to capacitive coupling between electrons inside and outside the cascade path.

In conclusion, we have demonstrated a cascade of electrons in a quantum dot array. We combined the cascade with Pauli spin blockade, and achieve spin readout fidelity potentially above 99.9% in 1.7 μs, even though the electrons were far from the charge sensor. We proposed that a cascade-based readout scheme will enable high-fidelity readout of spins in the interior of a two-dimensional quantum dot array, and that fanout of cascades can be used to enhance the signal further. Other platforms, for example, topological qubits, can also benefit from a cascade-based readout, when combined with parity-to-charge conversion[34]. The cascade of electrons opens up a new path for high-fidelity readout in large-scale quantum dot arrays, which is compatible with the established, high-sensitivity, charge sensor, paving the way for further progress in quantum computation and simulation with quantum dot arrays.

## Methods
**Device and set-up**. The material for the sample was grown with molecular beam epitaxy and consists of a GaAs/Al$_{0.3}$Ga$_{0.7}$As heterostructure with a silicon doping layer of density $7 \times 10^{12}$ cm$^{-2}$ at 50 nm depth from the surface. A 2DEG was formed at the interface, which is 90 nm below the surface. The mobility was $1.6 \times 10^6$ cm$^2$/Vs at an electron density of $1.9 \times 10^{10}$ cm$^{-2}$, measured at 4 K. A single layer of metallic gates (Ti/Au) is defined by electron-beam lithography. The gate pattern was designed to define eight quantum dots and two sensing dots. The device was cooled inside an Oxford Kelvinox 400HA dilution refrigerator to a base temperature of 45 mK. To reduce charge noise, the sample was cooled with bias voltages on the gates varying between 100 and 200 mV. Gates $P_1$, $P_2$, $P_3$, and $P_4$ were connected to bias-tees ($RC = 470$ ms), enabling application of a d.c. voltage as well as high-frequency voltage pulses. Voltage pulses were generated with a Tektronix AWG5014. The sensing dot resistance was probed with radio-frequency reflectometry. The $LC$ circuit for the reflectometry matched a carrier wave of frequency 97.2 MHz. The inductor, $L = 3.9$ μH, was a homebuilt, micro-fabricated NbTiN superconducting spiral inductor, and was wire-bonded to an ohmic contact. The reflected signal was amplified at 4 K with a Weinreb CITLF2 amplifier, and at room-temperature I/Q demodulated to baseband and filtered with a 10 MHz low-pass filter. Data acquisition was performed with a Spectrum M4i digitizer card. After digitisation, the I and Q components of the signal were combined with inverse-variance weighting.

**High-frequency voltage control**. The voltages for the charge-stability diagrams were simultaneously swept in 78 μs for the horizontal direction, and 6.2 ms for the

vertical direction. The signal was averaged over 1000 repetitions of such voltage scans. For the single-shot measurements, the voltage pulse durations were 100 μs, 100 μs, and 1 ms for, respectively, the empty, load, and read stage. After the read stage, a compensation stage of 1 ms was performed to prevent accumulation of charge on the bias-tees.

**Software**. The software modules used for data acquisition and processing were the open-source python packages QCoDeS, which is available at https://github.com/QCoDeS/Qcodes, and QTT, which is available at https://github.com/QuTech-Delft/qtt.

**Readout errors**. The error in the average charge readout fidelity, caused by relaxation during the arming time is estimated to be below $\eta_{arm} = \frac{1}{2}(1 - \exp(-t_{arm}/T_1)) = 0.015\%$. The error due to excitation during the arming and integration time is estimated to be below $\eta_{exc} = \frac{1}{2}(1 - \exp(-(t_{arm} + t_{int})/T_{exc})) = 0.014\%$. The mapping error in spin readout due to charge non-adiabaticity (and slow subsequent charge relaxation) can be estimated with the Landau-Zener formula to $\eta_{LZ} = \exp\left(-\frac{2\pi\alpha^2 \cdot \Delta t}{\hbar \Delta E}\right) = 10^{-9}\%$, with $\alpha = \sqrt{2}t_{c,12}$, and $t_{c,12} = 11.5$ μeV the tunnel coupling between dots 1 and 2, which is obtained from a spin funnel (see Supplementary Note 9), $\Delta t = 10$ ns is the ramp time of the pulse to the readout point and $\Delta E \approx 1$ meV is the change in double dot detuning from the load to the readout point. Note that charge noise can suppress the Landau-Zener transition probability, thus increasing the spin readout error owing to charge non-adiabaticity.

## Data availability
The data reported in this paper, and scripts to generate the figures, are archived at https://doi.org/10.5281/zenodo.3631337.

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

## Acknowledgements
The authors thank the members of the Vandersypen group for stimulating discussions. This work was supported by the Dutch Research Council (NWO Vici), and the Swiss National Science Foundation.

## Author contributions
C.J.v.D. performed the experiment and analysed the data, C.J.v.D. and T.-K.H. conceived the experiment and interpreted the data, U.M. fabricated the device, C.R. and W.W. grew the heterostructure, and C.J.v.D. and L.M.K.V. wrote the manuscript with comments from T.-K.H, and U.M.

## Competing interests
The authors C.J.v.D. and T.-K.H. are inventors on a patent application on cascade readout filed by Delft University of Technology (application no. 2024580). The remaining authors declare that they have no competing interests.
