## [Peer Review File · Nature Communications]

REVIEWER COMMENTS

Reviewer #1 (Remarks to the Author):

The paper demonstrates spin readout in a GaAs quantum dot circuit at a distance by means of a charge rearrangement process triggered by Pauli spin blockade. The data is very convincing and uncontroversial. It is presented well and the explanation is clear. The authors also speculate that this type of process could be utilized in more complex quantum circuits.

Such charge rearrangements have been known for over a decade, although in previous papers they are not referred to as 'cascades'. For example they are visible in almost every triple quantum stability diagram in neighboring charge configurations such as (0,1,0) to (1,0,1). What is novel in this paper is not the charge reconfiguration or 'cascade' but how it is triggered by the spin blockade process. The various additional electrostatic features related to enhancement etc.. are similar to those observed in latching readout schemes. This is certainly all a neat and clever idea and its demonstration is well done but I cant see how this rises to the level of a Nature Communication publication. So while this work will be of interest to the spin qubit community I cannot recommend its publication in Nature Communications.

Reviewer #2 (Remarks to the Author):

The paper electron cascade for spin readout describes a remote readout of the spin state of a singlet-triplet qubit. Pauli blockade has been used to distinguish spin states of a two-electron logical qubits encoded in the singlet or $m=0$ triplet state. A charge sensor detects whether two electrons can occupy the same QD signaling a singlet relative to a blockaded triplet. This paper introduces the observation that charge state can extend over multiple quantum dots and can produce a correlated response between remote QD sites allowing the Pauli-blockade transition to be sensed as a correlated response to the charge motion at the qubit location. The authors discuss co-tunneling and cascade versions of the effect, error mechanisms, make an estimate of the readout error in their particular instance and speculate on future micro-architectural implementations using this correlation effect that they coin as "electron cascade".

Readout of QDs embedded in 2D arrays is a potentially serious bottleneck for future QD quantum computing architectures. This paper addresses a timely and important challenge. The initial experimental results look very nice and are promising for the authors proposal. The device

architectural ideas are intriguing and very important for the community. But neither are sufficiently supported in the paper as it stands now. I don't support publication of this paper in Nature Communications without major revisions.

Required revisions should address the following:

(1) For the CPSB case with a 1.7 us readout time and a T1 of 680 us, one might estimate the relaxation error to go as $\exp(-t/T1) \sim t/T1 \sim 0.002$ or .2% which would not achieve 99.9% fidelity. Are the authors arguing that the T1 exposed time is a fraction of the total readout time? Only the first blockade step of the cascade? The error needs a more thorough discussion.

(2) The authors should really include evidence showing that the readout maps the S/T states to high/low signal with high fidelity. If the authors are going to claim that this is a 99.9% high fidelity readout, more evidence is necessary beyond showing a visibility histogram, as shown in figure 3. Visibility isn't necessarily an accurate mapping of the states to the readout signal. This reviewer appreciates that separation of state prep and measurement is a challenge in itself but it is still important for the authors to show that they have gone beyond assuming the mapping is correct based on a belief (put forth in this paper) that they understand the mechanism completely. I don't really doubt that the interpretation is correct, but for a high impact journal it's important to show that there is no doubt about the mapping particularly when arguing for 99.9% fidelity.

a. There are no units indicated in the T1 insets. Arb. Units?

(3) The charge stability diagrams do not show a great deal of change in signal contrast between methods for figure 2. Subsequent visibility in the histograms is undeniably better. Can the authors comment/clarify?

a. Perhaps related to this – is the factor of ~3 improvement an integrated improvement of ratio of shift in the instantaneous current from the charge sensor?

(4) The increase in signal that is observed in this paper is promising and the explanation is plausible, but can the authors rule out that the change in signal might be coming from electrostatic shift of the direction of the "dipole-like" shift of the charge or differences in screening when using the different charge state conditions?

(5) Why wasn't B-field applied to avoid triplet mixing and T1 effects? It would be more convincing to see this high fidelity achieved in a more conventional S/T operation mode with some B-field applied.

(6) An idealized Landau-Zener error might be estimated as a very low contribution as suggested in the paper ($1e-9$) but for a real system there are fast fluctuations from noise, as one source of alternate error. What is the Landau-Zener error when considering high frequency charge noise?

(7) In the appendix there is a discussion of average time in

the longer cascade (eqn. 7). Since the process is stochastic, there will be rarer tunneling events that take longer time. Shouldn't the time of the readout really be dominated by the longer time rare events and the error should include what fraction of the rare events you choose to discard and miss?

(8) The discussion of variation of Zeeman splitting along the path needs more clarification? Isn't the readout signal collapsed to a charge state at the PSB origin? Why does subsequent phase in a cascaded system matter?

(9) If the authors wish to make this paper about a new proposal for readout rather than focusing on the excellent readout results as they are, then they need to offer a clearer vision of why this concept is practically viable for scalable systems. It would be groundbreaking for extending and scaling, if it was clearer how this would really work for a 2D layout of many qubits and what the pros/cons are.

a. how does this readout really work for qubits in the interior of a dense 2D qubit array? Are there extra rows/columns of QDs inserted to pass the information? Are qubits shelved? Are there concerns about crosstalk. What (if any) extra idle time per qubit will be invoked to implement a scaled version of this readout?

b. an accounting of the extra QD resources required per qubit should also be discussed to provide a more balanced assessment of the pros and cons of this option (or solutions to these issues if the authors already have thought of this).

c. What happens if a charge gets stuck in a metastable state? Although infrequent, in large arrays extended by cascaded paths, it might occur that a charge is excited to a metastable condition and is 'stuck' out of place. See for example high frequency noise during Landau-Zener transition concern or excitation in to a shallow level, for example in silicon (valleys). Is this cascade readout practical in large systems with the additional complexity and failure modes? Have the authors estimated that these events are rare enough that they can be neglected from mention and that's why they are not discussed in this paper?

REVIEWER COMMENTS

Reviewer #1 (Remarks to the Author):

The paper demonstrates spin readout in a GaAs quantum dot circuit at a distance by means of a charge rearrangement process triggered by Pauli spin blockade. The data is very convincing and uncontroversial. It is presented well and the explanation is clear.

We thank the referee for the nice words on the data, presentation and explanation.

The authors also speculate that this type of process could be utilized in more complex quantum circuits. Such charge rearrangements have been known for over a decade, although in previous papers they are not referred to as 'cascades'. For example they are visible in almost every triple quantum stability diagram in neighboring charge configurations such as (0,1,0) to (1,0,1). What is novel in this paper is not the charge reconfiguration or 'cascade' but how it is triggered by the spin blockade process.

We agree with the referee that a process from a (0,1,0) charge state to a (1,0,1) charge state can be considered a short cascade. It is indeed the combination of a cascade with the Pauli spin blockade readout, which is one of the main novelties in our work. We also believe that showing a cascade in a quadruple dot (though with similar charge configurations as in a triple dot) makes the demonstration more compelling.

The various additional electrostatic features related to enhancement etc.. are similar to those observed in latching readout schemes.

The apparent similarity with latching is a signal enhancement as shown in Fig. 3 because in both schemes the movement of an electron within the quantum dot array is mapped to a change in total number of electrons on the quantum dot array.

However, there are important advantages of our cascade readout scheme compared to the latching scheme. The most important advantage of the cascade readout scheme is that it enables the readout over long distances, different from existing latching schemes. Furthermore, readout in large two-dimensional quantum dot arrays is an open challenge and a potential bottleneck for future QD quantum computing architectures, as also noted by Reviewer #2. Cascade based readout provides a novel approach to overcoming this challenge, whereas latching does not. In response to the comments of both referees, we have substantially rewritten the manuscript in order to bring out these advantages more strongly and also to support this claim with additional analysis.

For completeness, we note that an advantage of the latching scheme is that it may provide an increased relaxation time. In fact, cascade-based readout can be supplemented with latching to obtain the same benefit. Finally, latching requires a tedious tuning of tunnel couplings, whereas tuning of tunnel couplings in cascade based readout is more forgiving.

This is certainly all a neat and clever idea and its demonstration is well done but I cant see how this rises to the level of a Nature Communication publication. So while this work will be of interest to the spin qubit community I cannot recommend its publication in Nature Communications.

We hope that with these arguments and the corresponding substantial changes to the manuscript, the referee will appreciate the key step forward offered by cascade-based readout for large, and specifically two-dimensional quantum dot arrays, which is an important outstanding issue and will be of interest to the broad readership of Nature Communications.

Reviewer #2 (Remarks to the Author):

The paper electron cascade for spin readout describes a remote readout of the spin state of a singlet-triplet qubit. Pauli blockade has been used to distinguish spin states of a two-electron logical qubits encoded in the singlet or $m=0$ triplet state. A charge sensor detects whether two electrons can occupy the same QD signaling a singlet relative to a blockaded triplet. This paper introduces the observation that charge state can extend over multiple quantum dots and can produce a correlated response between remote QD sites allowing the Pauli-blockade transition to be sensed as a correlated response to the charge motion at the qubit location. The authors discuss co-tunneling and cascade versions of the effect, error mechanisms, make an estimate of the readout error in their particular instance and speculate on future micro-architectural implementations using this correlation effect that they coin as “electron cascade”.

Readout of QDs embedded in 2D arrays is a potentially serious bottleneck for future QD quantum computing architectures. This paper addresses a timely and important challenge. The initial experimental results look very nice and are promising for the authors proposal. The device architectural ideas are intriguing and very important for the community. But neither are sufficiently supported in the paper as it stands now. I don't support publication of this paper in Nature Communications without major revisions.

We would like to thank the referee for the compliments on the experimental results and ideas for quantum dot computing architectures. We intended to present our work as a novel scheme for readout in larger quantum dot arrays supported by a state-of-art demonstration of readout with such scheme. With our revisions and responses we have improved the focus of our work to give more support to our ideas on cascade-based readout in future quantum dot architectures and present the readout results as a promising proof-of-principle demonstration for these ideas. We have provided more detail about the implementation in future quantum dot architectures, but have tried to keep a balance, as there are many different implementations imaginable which could benefit from cascade-based readout, and going into too much detail for one such implementation would draw away attention from the more general applicability of cascade-based readout. We hope the referee appreciates our revisions of the manuscript and our choice for renewed and improved focus of the manuscript, and this will convince the referee that our work on cascade-based readout is suitable for publication in Nature Communications.

Required revisions should address the following:

(1) For the CPSB case with a 1.7 μs readout time and a T_1 of 680 μs , one might estimate the relaxation error to go as $\exp(-t/T_1) \sim t/T_1 \sim 0.002$ or .2% which would not achieve 99.9% fidelity. Are the authors arguing that the T_1 exposed time is a fraction of the total readout time? Only the first blockade step of the cascade? The error needs a more thorough discussion.

*We agree that using $\exp(-t/T1)$ and approximating this exponent with $t/T1$ can be useful to get a rough estimate of the readout error, though this first estimate results in a too pessimistic error. There are two main differences with our error analysis, which explain why the error due to relaxation during the integration time contributes only an error of 0.068 %, which is the value reported in the main text. First, the estimate of the referee $\exp(-t/T1)$ corresponds to the probability that a triplet state will decay to a singlet state somewhere during the time interval t . But, this is not the same probability as that a triplet state is detected as a singlet state, because the readout signal is integrated over the entire interval. Imagine that the decay of the triplet occurs at the end of the interval, then during almost all of the integration time the signal would correspond to a triplet state, hence the integrated signal will be very close to that for a triplet. The relaxation during the integration time is taken into account by the fitting of the histogram, see [Barthel, PRL (2009)], or also [Gambetta, PRA (2007)]. Thus using as estimate $\exp(-t/(2*T1)) \sim t/(2*T1)$ would result in a more realistic error. Second, the estimate the referee describes is an estimate of the error in the triplet readout, while the fidelity of 99,9% we report is the average readout fidelity, i.e. averaged over the singlet and triplet readout fidelities, thus the triplet readout error contributes only half that error to the average readout error. To clarify this in our work we have added Section III to the Supplementary information with the model used for obtaining the error due to residual overlap of the probability density distributions and relaxation during the integration time.*

(2) The authors should really include evidence showing that the readout maps the S/T states to high/low signal with high fidelity. If the authors are going to claim that this is a 99.9% high fidelity readout, more evidence is necessary beyond showing a visibility histogram, as shown in figure 3. Visibility isn't necessarily an accurate mapping of the states to the readout signal. This reviewer appreciates that separation of state prep and measurement is a challenge in itself but it is still important for the authors to show that they have gone beyond assuming the mapping is correct based on a belief (put forth in this paper) that they understand the mechanism completely. I don't really doubt that the interpretation is correct, but for a high impact journal it's important to show that there is no doubt about the mapping particularly when arguing for 99.9% fidelity.

We agree with the referee that mapping of spin states to readout signals is relevant for qubit readout. But as we aim to focus the paper on the novel possibilities offered by cascade readout, we believe that such error analysis or additional measurements to make estimates of such errors do not fit within the scope of our work. To lift some of the focus on the high-fidelity spin readout we have rewritten the corresponding part of the paper as we believe that in this way we shift the focus of our work more to the novelty of the scheme and its relevance for future architectures. We have adjusted the main text to clarify that our spin readout fidelity does not include errors due to mapping from the (1101) spin states to the measurement basis at the readout point, and have also adjusted the introduction and conclusion accordingly.

Supplementary Note VIII does describe considerations for the mapping of spin states from the loading point to the measurement basis at the readout point for our prototype of cascade-based readout. We would like to note that mapping error caused by the nuclear spins in our material can be suppressed by performing dynamic nuclear polarization with feedback to suppress nuclear gradient field fluctuations or by using purified Si, as mentioned in Supplementary Note VIII. We aim to demonstrate that high fidelity spin readout is achievable with a cascade-based scheme, which we believe is supported by the experimental results and error analysis in the main text.

a. There are no units indicated in the T1 insets. Arb. Units?

The authors would like to thank the referee for pointing this out. We have added a comment in the caption of the relevant figures on the units for the vertical axis of the insets, which are indeed the same units as on the horizontal axis of the histogram.

(3) The charge stability diagrams do not show a great deal of change in signal contrast between methods for figure 2. Subsequent visibility in the histograms is undeniably better. Can the authors comment/clarify?

For the charge stability diagrams in Fig. 2b the sensor peak position was intentionally slightly shifted by changing the sensing dot plunger voltage, such that all charge states would result in clearly different sensor signals, but with this shift the sensor is not optimized for the Pauli spin blockade readout. The shift of the sensor is explained in Supplementary Note 1. In Fig. 2c a charge stability diagram is shown for which the sensor was tuned similar to how it was tuned for the single-shot histograms of Fig. 3.

a. Perhaps related to this – is the factor of ~3 improvement an integrated improvement of ratio of shift in the instantaneous current from the charge sensor?

The signal enhancements are obtained from the single-shot readout data. It is the ratio of the signal-to-noise ratio from demodulated RF signal as reflected from the sensing dot. The signals were integrated over 1.5 μ s.

(4) The increase in signal that is observed in this paper is promising and the explanation is plausible, but can the authors rule out that the change in signal might be coming from electrostatic shift of the direction of the “dipole-like” shift of the charge or differences in screening when using the different charge state conditions?

We appreciate this point by the referee, and agree that such effects could cause a signal difference, but we are convinced that these effects do not cause the signal enhancement we observed. The difference in sensor signals between charge states (1100) and (0200) is very similar to that between (1101) and (0201), while the difference is much larger between (1101) and (0200), thus any difference in direction of the dipole-like shift of the electron on the two leftmost dots due to the additional electron on the rightmost dot does not have a significant effect on the differences in sensor signal, and similarly for any change in screening effects due to the additional electron on the rightmost dot. Also the charge-stability data in Fig. S1b shows the same behaviour when comparing the differences between (1101) and (0201), and (1110) and (0210) to the difference in sensor signal between (1110) and (0201). We have added a comment to the main text.

(5) Why wasn't B-field applied to avoid triplet mixing and T1 effects? It would be more convincing to see this high fidelity achieved in a more conventional S/T operation mode with some B-field applied.

We agree that for singlet-triplet qubit readout an external B-field would be applied, and that this influences the T1 and mixing effects. Since we have shifted the focus away from high-fidelity spin readout in the revision, we believe this point is less critical. Furthermore, and importantly, Pauli spin blockade

readout is of more general interest, which includes cases without external B-field, one example is the recent work from our group: “Nagaoka ferromagnetism observed in a quantum dot plaquette.”.

(6) An idealized Landau-Zener error might be estimated as a very low contribution as suggested in the paper (1e-9) but for a real system there are fast fluctuations from noise, as one source of alternate error. What is the Landau-Zener error when considering high frequency charge noise?

We appreciate this comment by the referee, and indeed the Landau-Zener transition probabilities depend on the noise in the system. However, any errors as estimated from a Landau-Zener analysis for our system will be suppressed, because of rapid charge relaxation. For the tunnel coupling value the experiment is performed at, the charge relaxation rate is estimated to be 0.1-1 GHz at the readout point, and is even > 10 GHz along the ramp to the read-out point, when closer to the inter-dot transition [see Barthel, PRB (2012)]. During the 10 ns ramp from the loading point to the readout point, and during the first nanoseconds of the arming time of 200 ns, any excited charge state rapidly decays, and would have negligible effect on the readout signal. In other materials and for smaller tunnel coupling values the charge relaxation could be slower. See our response to point 9c for further comments on errors due to occupying an excited charge state.

We have adjusted the section about error analysis in the main text as mentioned in our response for point 2. In this adjustment we have added a comment to warn about the effect of high-frequent charge noise on the reliability of the estimate based on the LZ formula.

(7) In the appendix there is a discussion of average time in the longer cascade (eqn. 7). Since the process is stochastic, there will be rarer tunneling events that take longer time. Shouldn't the time of the readout really be dominated by the longer time rare events and the error should include what fraction of the rare events you choose to discard and miss?

We appreciate that the referee raises this point as it has allowed us to improve our work. The longer-time rare events can indeed result in readout errors. Equation (7) describes the average duration for the sequential implementation, in which the movement of each, except the first, of the electrons is conditioned on the movement of the previous electron. To analyse the readout error due to longer-time rare events the probability distribution of the durations is important. The conditional character, assuming homogeneous tunnel rates, results in an Erlang/Gamma distribution of the duration for the cascade, which as expected indicates a sublinear scaling for longer time events with respect to the cascade length. We have added subsection V.E. in the supplementary information to describe this analysis in detail. To optimize the readout fidelity with respect to long time events the distribution of the duration could be taken into account for the integration of the sensor signal. We also note that in the controlled propagation implementation such latching errors would be suppressed by the adiabatic character of each of the transitions, and in the cases a charge still latches the subsequent rapid charge relaxation would further suppress an error [See also our response to point (6)].

(8) The discussion of variation of Zeeman splitting along the path needs more clarification? Isn't the readout signal collapsed to a charge state at the PSB origin? Why does subsequent phase in a cascaded system matter?

The readout indeed collapses the charge state at the PSB origin. However, the spins of the electrons residing within the cascade path are moved conditioned on the outcome of the readout. If such an electron moves between dots with different Zeeman splitting, then the spin of the electron will pick up a phase relative to the original Zeeman splitting. This phase is relevant when the electrons along the cascade path themselves are used to store quantum information. Such phase can be compensated in software [T.F. Watson, Nature (2018)] or with hardware provided it is known.

Uncertainty in the electron position can result in an unknown phase error. If the cascade is implemented sequentially, then uncertainties for electrons in the cascade path add up for the electrons further down the cascade path, thus such scheme is more susceptible to phase errors, especially as the cascade path becomes longer. If the cascade is implemented adiabatically, then the electron position can be accurately tracked, and any phase errors will be small, but the cascade should be operated rather slowly to remain adiabatic for a long cascade, due to the co-tunnel character as described in Supplementary Note IV. A. and B. By implementing the cascade in the controlled propagation scheme as described in Supplementary Note IV. C., the adiabatic character of the cascade can be preserved, while also largely maintaining the speed, as the total duration of the cascade increases linearly with the cascade length.

(9) If the authors wish to make this paper about a new proposal for readout rather than focusing on the excellent readout results as they are, then they need to offer a clearer vision of why this concept is practically viable for scalable systems. It would be groundbreaking for extending and scaling, if it was clearer how this would really work for a 2D layout of many qubits and what the pros/cons are.

We thank the referee for the compliment on the readout results and appreciate the questions about the implementation in future quantum dot-based architectures. Before going into more detail on a specific layout of quantum dots/qubits, we would like to emphasize that the cascade-based readout can be implemented in and be useful for various layouts. The cascade-based readout requires as ingredients Coulomb repulsion between electrons, tunnel coupling to enable the motion of electrons, and control over dot chemical potentials, which each are very generally present and required for quantum dot computing architectures. The cascade-based readout could be implemented with some empty dots to facilitate movement of electrons, thus dots those cannot host a qubit, but empty dots are not required.

To further illustrate that cascade-based readout can be useful for various layouts, let us consider another example: a (long) 1D array. Currently, near each ~ 3 dots a sensing dot is placed for readout, continuing that approach would result in many sensors for longer 1D arrays. The many sensors would require many electron reservoirs, and many inductors for RF readout. With cascade-based readout a single sensor could be used to read-out many qubits, thus the requirements for chip design and additional electronics would be greatly reduced.

a. how does this readout really work for qubits in the interior of a dense 2D qubit array? Are there extra rows/columns of QDs inserted to pass the information? Are qubits shelved? Are there concerns about crosstalk. What (if any) extra idle time per qubit will be invoked to implement a scaled version of this readout?

We imagine an implementation without dedicated rows/columns for information transfer nor qubit shelving, but also do not exclude that an implementation which includes these could eventually prove to be preferable.

The electrons within the cascade path can be operated as qubits as discussed in the main text, but during the cascade readout these qubits may move, thus they are not available for gate operations during the readout. A qubit outside but near the cascade path indeed can experience capacitive crosstalk, which will not move the corresponding electron because it is deep in Coulomb blockade but it may influence the qubit state, albeit less than for qubits within the cascade path. The influences can be accounted for similarly to the effects of motion of the qubits within the cascade path. Qubits outside the path which do not suffer from crosstalk can be operated during the cascade-based readout, thus are not kept idle. We like to note that care must be taken in tracking and handling phase shifts and (artificial or natural) spin-orbit induced rotations. We have added a comment in the main text to discuss that capacitive coupling between electrons inside and outside the cascade path can cause correlated errors, and extra caution should be taken to handle these.

b. an accounting of the extra QD resources required per qubit should also be discussed to provide a more balanced assessment of the pros and cons of this option (or solutions to these issues if the authors already have thought of this).

One implementation is a checkerboard filling of a two-dimensional array of quantum dots with single electron spin-qubits as illustrated in Fig. 4a. The cascade path is formed of dots for which chemical potentials are tuned close to a charge transition, while electrons on dots outside the cascade path are deep in Coulomb blockade. The downside of the checkerboard filling is that N dots only host $N/2$ qubits, but because of the small dimensions of quantum dots, this filling still allows for very dense packing of qubits.

Other implementations could entail higher fillings, as illustrated with the added one-dimensional example in Fig. 4d and described in the main text. In this implementation the empty dot, needed for the cascade, is emptied by the PSB readout itself. In this scheme the electrons in the cascade move away from the charge sensor. In this implementation the voltage pulse for PSB is first performed, and subsequently a voltage pulse is sent to tune the dot array to activate the cascade. This two-step procedure suppresses co-tunnelling from (111...) to (210...) without occupying (201...).

c. What happens if a charge gets stuck in a metastable state? Although infrequent, in large arrays extended by cascaded paths, it might occur that a charge is excited to a metastable condition and is 'stuck' out of place. See for example high frequency noise during Landau-Zener transition concern or excitation in to a shallow level, for example in silicon (valleys). Is this cascade readout practical in large systems with the additional complexity and failure modes? Have the authors estimated that these events are rare enough that they can be neglected from mention and that's why they are not discussed in this paper?

If a charge within the cascade path gets stuck, then this could result in a readout error, depending on how long the charge remains stuck. In the newly added Supplementary Note V.E. we discuss scaling considerations for errors due to latching. Without going into much more detail, it is hard to give quantitative estimates for future devices, as the probabilities to get and remain stuck depend on many parameters, which vary with device tuning and material, making such analysis beyond the scope of our work. But we have added section V.E. in the supplementary which discusses the scaling of latching in the cascade with respect to the cascade length.

REVIEWER COMMENTS

Reviewer #2 (Remarks to the Author):

The authors have adjusted the manuscript in response to both referees comments. The authors have resolved the concerns indicated in those earlier reports.

Overall the paper shows a proof of principle of remote readout using what they call a cascade mechanism. The demonstration indicates the potential for high fidelity. The authors further illustrate conceptual layouts in which the remote readout might be used. This work is very timely and is high impact.

The paper seems ready for publication except for one request.

Please modify remaining claims of high fidelity as something like 'potential for high fidelity' or some other agreeable modifier. The authors are forthcoming about the readout fidelity estimate not accounting for all forms of error. So it seems that it is more accurate to claim 'potential for high fidelity', in places such as the abstract.

Reviewer #3 (Remarks to the Author):

The paper demonstrates 'electron cascade' as a new readout technique for spin qubits in large quantum dot arrays. High-fidelity spin readout was emphasized in the previous manuscript, but the authors revised the manuscript to focus more on the capability of reading out distant spin qubits in

large 1D/2D arrangements. This change of the focus is fair because the high-fidelity feature is similar to the latching effect as suggested by Reviewer #1 and the authors would need additional analyses to convince the high fidelity spin readout as Reviewer #2 suggests in his/her comments (2,5). I agree with the view of Reviewer #2 that the revised manuscript should provide a clear vision of how the electron cascade is viable for larger systems. While the manuscript partially addresses possible challenges in such applications, I'm not convinced that the proposed technique is viable nor advantageous compared to existing proposals. Considering the high standard of Nature Communications, I cannot recommend publication of the paper.

Readout of distant spin qubits is an undoubtedly important challenge for future quantum computing architectures. The authors partially address comments (9a-9c) raised by Reviewer #2, but I think the following points need to be addressed more thoroughly to enable fair comparison with other techniques such as spin shuttling [T. Baart et al., Nat. nano. 11 330] and resonant SWAP [A. Sigillito et al., npj Quantum Information 5 110].

1.

The authors describe possible 2D layouts in Fig. 4, but I cannot see what is really the potential advantage of the cascade readout. For instance, is the cascade readout easier than using spin shuttling to bring spin qubits to near the charge sensor? The authors argue that each electron along a cascade path can still be operated as a spin qubit, but this is true only in the 'controlled propagation' mode that is not demonstrated in this paper.

2.

Could the authors explain how the operation window scales in dense 1D/2D arrays to claim that the cascade readout is viable in those systems? Is it feasible to have a sequential transition such as (111010...)->(021010...)->(020110...)->(020101...)->... in a long array?

3.

I think that the risk of a charge stuck out of place (in some excited states) suggested by Reviewer #2 (9c) could be really serious. I understand that quantitative estimates are out of the scope, but scaling consideration like the one for the cascade speed should be provided.

REVIEWER COMMENTS

Reviewer #2 (Remarks to the Author):

The authors have adjusted the manuscript in response to both referees comments. The authors have resolved the concerns indicated in those earlier reports.

Overall the paper shows a proof of principle of remote readout using what they call a cascade mechanism. The demonstration indicates the potential for high fidelity. The authors further illustrate conceptual layouts in which the remote readout might be used. This work is very timely and is high impact.

The paper seems ready for publication except for one request.

Please modify remaining claims of high fidelity as something like 'potential for high fidelity' or some other agreeable modifier. The authors are forthcoming about the readout fidelity estimate not accounting for all forms of error. So it seems that it is more accurate to claim 'potential for high fidelity', in places such as the abstract.

We are grateful for the referee's comments about our work and revision, and are pleased to read that the referee finds our work almost ready for publication in Nature Communication. Following the suggestion by the referee, we have adjusted the wording of the high fidelity claims, in line with our results, and hope that with this adjustment the referee will find our work ready for publication in Nature Communications.

Reviewer #3 (Remarks to the Author):

The paper demonstrates 'electron cascade' as a new readout technique for spin qubits in large quantum dot arrays. High-fidelity spin readout was emphasized in the previous manuscript, but the authors revised the manuscript to focus more on the capability of reading out distant spin qubits in large 1D/2D arrangements. This change of the focus is fair because the high-fidelity feature is similar to the latching effect as suggested by Reviewer #1 and the authors would need additional analyses to convince the high fidelity spin readout as Reviewer #2 suggests in his/her comments (2,5). I agree with the view of Reviewer #2 that the revised manuscript should provide a clear vision of how the electron cascade is viable for larger systems. While the manuscript partially addresses possible challenges in such applications, I'm not convinced that the proposed technique is viable nor advantageous compared to existing proposals. Considering the high standard of Nature Communications, I cannot recommend publication of the paper.

We appreciate the referee's comments on our work and revision.

We want to emphasize that the signal enhancement of our demonstration is different in origin than the latched readout. With latched readout an inter-dot transition of a double dot is combined with a dot-reservoir transition of one of the two dots, while with the cascade scheme a transition between some dot(s) results in transitions between other dot(s) far away. Thus the latching scheme offers a signal enhancement for the readout of spins near a sensor, while the cascade scheme enables the readout of distant spins, i.e. far away from the sensor.

We address the viability and advantage of cascade-based readout compared to existing proposals through our responses to the questions below.

Readout of distant spin qubits is an undoubtedly important challenge for future quantum computing architectures. The authors partially address comments (9a-9c) raised by Reviewer #2, but I think the following points need to be addressed more thoroughly to enable fair comparison with other techniques such as spin shuttling [T. Baart et al., Nat. nano. 11 330] and resonant SWAP [A. Sigillito et al., npj Quantum Information 5 110].

1.

The authors describe possible 2D layouts in Fig. 4, but I cannot see what is really the potential advantage of the cascade readout. For instance, is the cascade readout easier than using spin shuttling to bring spin qubits to near the charge sensor? The authors argue that each electron along a cascade path can still be operated as a spin qubit, but this is true only in the 'controlled propagation' mode that is not demonstrated in this paper.

We value that the referee raises this relevant point. Below we have listed several advantages of the cascade-based readout compared with readout based on shuttling or gate operations.

- For shuttling schemes as described in [T. Baart et al., Nat. nano. 11 330], all electrons in the shuttling path are pushed into the reservoir one after another, hence their quantum information cannot be maintained. In contrast, in cascade-based readout the dots in the cascade path can host spin qubits for which the state can be maintained. Alternatively, shuttling could be implemented through series of empty dots forming the shuttling paths, but this brings overhead and breaks the qubit-qubit connectivity. This is not the case for the cascade-based readout.

- For shuttling, one or multiple electrons are moved over long distances while with the cascade many electrons on the cascade path are moved over very small distances. This offers benefits as spin qubits do not have to be moved out of their local environment, and the connectivity of qubits can be approximately preserved. This is beneficial for quantum error correction where error propagation must be minimized.

- Fanout is not possible with shuttling but it is with cascades.

- If a spin state is transferred via a series of gate operations, e.g. resonant SWAP operations as in [A. Sigillito et al., npj Quantum Information 5 110], then the readout fidelity becomes limited by the product of an entire chain of two-qubit gate fidelities.

We have included a summarized version of the above comments including references into the introduction of the main text of the paper as we agree with the referee that this comparison is of interest.

Finally, even though the controlled propagation mode is not the focus of the present experiments, it is not a speculative idea since controlled propagation is routinely implemented in double-dot experiments.

2.

Could the authors explain how the operation window scales in dense 1D/2D arrays to claim that the cascade readout is viable in those systems?

For the first step, which is the pulse for the PSB, the operating window is the same as for conventional PSB readout with all other dots occupied and in Coulomb blockade. For the second step, which activates the cascade in a dense array, the operating window is dominated by the on-site interaction, which prevents double occupation on the dots in the cascade path, thus resulting in a relatively large operating window for those dots. We have added additional comments on considerations about the implementation for dense arrays in the main text.

Is it feasible to have a sequential transition such as (111010...)->(021010...)->(020110...)->(020101...)->... in a long array?

Such a sequential transition is feasible, because interaction strengths decay with distance, thus the size of the operating window only depends on the occupation of nearby dots, and similarly for the operating window for the other pairs in the transition. In Supplementary Note II.C. we derive the operating window for what we refer to as the “cascade with inter-dot”, which is the smaller-scale version of the long array as described by the referee. We have added a comment to Supplementary Note II analogous to this explanation.

3.

I think that the risk of a charge stuck out of place (in some excited states) suggested by Reviewer #2 (9c) could be really serious. I understand that quantitative estimates are out of the scope, but scaling consideration like the one for the cascade speed should be provided.

We agree with the referee that the probability of charge to get stuck is a relevant factor to consider. In Supplementary Note V.E. in the supplementary information we deduce that the time for a cascade to complete with a given probability scales sub-linearly with the cascade length. Thus the scaling for the probability for a charge to get stuck, also scales sub-linearly with cascade length. We have added a comment to the specific section to clarify the above, and have adjusted the heading of Supplementary Note V to improve clarity.